# Evaluation of Headache Trends among Undergraduate First Responders for Medical Emergencies at Saudi University in Riyadh, Saudi Arabia

**DOI:** 10.3390/medicina59091522

**Published:** 2023-08-23

**Authors:** Abdullah M. Alobaid, Wajid Syed, Mahmood Basil A. Al-Rawi

**Affiliations:** 1Department of Accident and Trauma, Prince Sultan bin Abdulaziz College for Emergency Medical Services, King Saud University, Riyadh 11451, Saudi Arabia; 2Department of Clinical Pharmacy, College of Pharmacy, King Saud University, Riyadh 11451, Saudi Arabia; 3Department of Optometry, College of Applied Medical Sciences, King Saud University, Riyadh 11451, Saudi Arabia; basilalravi@gmail.com

**Keywords:** headache, intensity, photophobia, photophobia, analgesics

## Abstract

*Background and objective*: Headache is more common among students and may be a crucial indication of mental health; it can have a detrimental impact if left untreated, especially on students, and may affect their academic performance. Thus, this study aimed to assess the headache trends among undergraduate first responders for medical emergencies at Saudi University in Riyadh, Saudi Arabia. *Method*: A cross-sectional study was undertaken over three months in 2023 to analyze headache patterns among emergency medical services (EMS) students using prevalidated questionnaires. The data were collected via convenience sampling and processed with the IBM SPSS Statistics 26 program (IBM Inc., Chicago, IL, USA). *Results*: The current findings show that the majority 90.6% (*n* = 164) of the participants had at least one episode of headache in the last week. With regard to the number of days students had a headache during the last week, 21.5% (*n* = 39) of them reported 2 days, while 18.8% (*n* = 34) reported 1 day only. About slightly less than half (48.6% (*n* = 88)) of students reported that the usual site of the headache was frontal, followed by temporal 27.1% (*n* = 49), and 23.3% (*n* = 42) reported orbital. When students were asked whether a headache aggravates their routine activity, 40.3% agreed with this statement, and 44.2% of students reported that a headache causes them to avoid their routine activity (reading writing, attending college). The majority of the students considered taking analgesics (60.8%), followed by sleeping (26.5%), caffeine (14.4%), and herbal and alternative remedies (7.7%) for the relief of headaches. Although headache episodes were significantly associated with routine activities (reading writing, attending college) (*p* = 0.018, *t* = 2.282) daily activities (walking, running) (*p* = 0.022, *t* = 2.307). The findings showed that fourth-year EMS students were found to have higher headache pain intensity scores compared to other students (*p* = 0.046). Similarly, the pain intensity was significantly higher among the students between 1–3 and >7 episodes of headaches (*p* = 0.001) *Conclusion*: The findings of this study revealed that Saudi emergency medical services students suffer from headaches. However, the current findings revealed variation in the headache pain intensity scores concerning the year of study.

## 1. Introduction

The World Health Organization (WHO) reported that over half of the adult population had a headache in the preceding year, yet the afflicted individuals never went to a healthcare facility for seeking headache treatment, instead preferring to manage their headaches themselves using alternative treatments [1,2]. Earlier studies reported that headache is more prevalent among university students in comparison to other populations [3,4,5,6,7,8,9]. Headache is a very common neurological disorder that affects people of all ages and genders. It is associated with personal and social difficulties such as discomfort, incapacity, decreased quality of life, and financial implications [1]. The incidence and prevalence of headaches are significantly different concerning population and headache types [5,6,7,8]. According to recent estimates, 75% of adults around the world suffer from headaches, and primary headaches are the most common public health concern that goes untreated [3]. Adults in the United States had a prevalence of 15.9%, whereas adults in the United Kingdom had a prevalence of 15% [4,5]. According to a recent assessment in a small region of Saudi Arabia, 3% of adults suffer from headaches [6]. On the contrary, data about students revealed a higher prevalence of headaches [7,8,9,10]. For instance, the prevalence of headaches among Saudi medical students was 53.8% [7], and among female students, it was 68.4% (Desouky et al., 2019) [8]. Among Ethiopian undergraduates, it was 34% [9]. A 73.1% prevalence was found in a recent study by Panigrahi et al. among Indian students [10]. On the other hand, the prevalence was very high, 96%, among Omani medical students [11]. Similar findings were found in recent research on pharmacy nursing students, showing that 66.1% of the participants experienced at least one episode of headache [12]. There was evidence that the prevalence of headaches among healthcare students in comparison to other categories of students is much higher, with an estimate of 91% [13,14,15,16]. Additionally, studies nationally and internationally reported that undergraduates suffer from a high frequency of headaches [7,8,9,10,11,12].

Despite being very common, headaches interrupt students’ academic behavior and workplace environment, and limitations in academic activities result in lower performance in their student life [17,18,19]. The incidence of headaches was increasingly more common among young adults [20]. The most common causes of headaches among students were exposure to extreme stress, which was related to curriculum, examinations, study load, sleeping issues, lifestyle behaviors such as smoking, excessive coffee/soft drink intake, family history, or genetics [21,22,23]. Other studies revealed that greater use of digital technology is connected with an increased risk of headache, as is prolonged computer usage and excessive (>4 h/day) use of video games among teenagers [4,5]. Furthermore, increased smartphone use has been connected to headache, sleep disruption, cognitive impairment, and weariness, with call frequency being strongly tied to headache risk [6].

Over the past several decades, both nationally and internationally, the number of patients visiting emergency departments (EDs) has constantly increased [24,25]. Because of how they care for patients in difficult circumstances, EMS professionals are in more demand every day. In emergencies or at the time of any disaster, they are required to act quickly and efficiently to save lives and minimize injuries. Natural and human-made disasters are becoming more common around the world, resulting in loss of life and property as well as infrastructure damage [24,25]. Today students are tomorrow’s professionals. It is well known that stress, the difference in working shifts, and workload are more common among EMS personnel, which increases the chances of having headaches, compared to other professionals or students, which raises the issues of mental disturbances among first responders for emergencies, since they are the healthcare providers in difficult situations or emergencies. Furthermore, EMS students are required to perform their activities in an emergency promptly, have longer class or practical hours, and have high levels of stress, which may impair their academic performance and general quality of life. A few studies have been conducted on headache disorders among medical, pharmacy, and healthcare students [7,8,9,10,11,12], but no study has examined headache disorders among EMS students. Therefore, this study aimed to assess the headache trends among healthcare students at a university in Riyadh, Saudi Arabia.

## 2. Methods

### 2.1. Design and Ethical Aspects

A cross-sectional questionnaire-based study was performed among emergency medical service (EMS) students at King Saud University’s Prince Sultan College for Emergency Medical Services in Riyadh, Saudi Arabia, from March to May 2023. The study was conducted in accordance with the Declaration of Helsinki and approved by the Institutional Review Board of King Saud University (protocol code E-23-8027) College of Medicine, Riyadh, Saudi Arabia

### 2.2. Inclusion and Exclusion Criteria

The target population consisted of all students over the age of 18 who commonly suffered from headaches, could speak Arabic or English, and agreed to participate in the study by signing a consent form. Students with headaches caused by viruses, hangovers, colds, or head injuries, as well as those who did not meet the inclusion requirements, were eliminated from the study. The participants were informed that the data would only be used for research purposes and would be kept anonymous during the study. Others who did not meet the inclusion criteria were ruled out.

### 2.3. Sample Size

The necessary sample size was determined using a computerized sample size calculator (http://www.raosoft.com/samplesize.html. accessed on 1 June 2023) that has been used in an array of prior studies [26,27,28,29,30,31,32,33,34,35]. On the KSU campus, in the College of EMS, there were about 300 students from whom the required sample size was obtained at a 95% confidence interval (CI) and a 5% margin of error (ME). To calculate the sample size, we used a 5% non-response rate, which led to a sample size of 194. Nevertheless, we ultimately included 200 students in our study to increase the reliability.

### 2.4. Questionnaire Design and Data Collection

The questionnaire intended to assess headaches and their trends among EMS students was adapted from similar studies [12]. The questionnaire was broken into three pieces. The first component included demographic and clinical measures that assessed participants’ characteristics such as age, gender, and information about their headaches. The second section included six items about the characteristics and patterns of headaches, such as the location of the headache, headache and routine activity, and symptoms that are accompanied or followed by headache. The final section discussed the severity of pain and other symptoms related to headaches, as well as management strategies (11 items). All of these questions were graded using binary responses, continuous scales, and multiple-choice answers.

The questionnaire draft was initially created in English, and a research team analyzed it to ensure that the text was clear and accurate. To ensure the readability and simplicity of administration of the questionnaires, a pilot study was conducted among a randomly chosen small sample of students (*n* = 30). The outcomes of the pilot investigation were excluded from the final analysis. Using Cronbach’s alpha, the questionnaire’s reliability was assessed, and the result was 0.79, indicating that the questionnaire was reliable for use in the study. The final study questionnaire was delivered by personally visiting students who were currently enrolled and present at the time of data collection each year. A team of researchers trailed students to collect as many responses as possible. To eliminate duplicate responses, participants were advised to complete the questionnaire just once, regardless of how they obtained it.

### 2.5. Data Analysis

The data were analyzed with the Statistical Package for the Social Sciences (SPSS) (version 26 for Windows) (SPSS Inc., Chicago, IL, USA). The demographic and clinical variables were combined using frequencies (n) and percentiles (%), the mean (M), and the standard deviation (STD). To assess the differences between the variables at a 95% confidence interval (CI), and with *p* values of 0.05 indicating a significant difference, to compare means for two groups and more than two groups, independent t-tests and one-way ANOVA tests were utilized.

## 3. Results

### 3.1. Demographic and Clinical Characteristics

A total of 200 students were recruited for the study, and 181 of them completed the questionnaires, yielding a response rate of 90.5% (Figure 1). Ninety-five percent (*n* = 172) of the participants were between the ages of 18 and 22. The second-year students dominated with 38.7% (*n* = 70), followed by the third-year students with 35.4% (*n* = 64) and the fourth-year students with 17.1% (*n* = 31). The mean age of the students who suffered from headaches was 20.4 (SD = 1.90). The current findings show that the majority (*n* = 164; 90.6%) of the participants had at least one episode of headache in the last week. With regard to the number of days students had a headache during the last week, 21.5% (*n* = 39) of them reported 2 days, while 18.8% (*n* = 34) reported 1 day only. On the other hand, 74%(*n* = 134) of the participants reported 1 to 3 episodes of headache per day. Furthermore, 56.6% (*n* = 103) of the students reported 5–6 h, while 14.3% (*n* = 26) of the students reported that their headache lasted between 1 and 4 h. On the other hand, 18.1% (*n* = 33) of the students reported between 40 and 60 min, and only 1.6% (*n* = 3) of them reported 7–30 min (<30 min). Students’ frequencies for the demographic characteristics are shown in Table 1.

### 3.2. Characteristics of Headaches among Participants

About slightly less than half (48.6% (*n* = 88)) of students reported that the usual site of the headache was frontal, followed by temporal 27.1% (*n* = 49), and 23.3% (*n* = 42) reported orbital, although two-thirds of the students (60.8% (*n* = 110)) reported as bilateral while more than one-third (39.2% (*n* = 71)) reported a unilateral type of headache. Usually, 52.5% (*n* = 95) reported headache quality as pressing/tightening, while 40.9% (*n* = 74) described it as throbbing/pulsating. When students were asked whether a headache aggravates their routine activity, 40.3% agreed with this statement, and 44.2% of students reported that a headache causes them to avoid their routine activity. Detailed information is illustrated in Table 2.

For instance, students reported that headaches were usually associated with photophobia; 33.7% revealed that they were sensitive to light, and only 27.6% revealed that the headaches were phonophobic. More than one-third (38.1%) of the students had a family history of headaches. Only 13.3% of students sought medical attention for headaches; in this study, 22% of undergraduate EMS students reported that they did not have any aura symptoms, while 13.2% had wavy lines or spots as aura symptoms. Detailed information about aura symptoms is illustrated in Figure 2.

In this study, one hundred sixteen (61.1%) students reported that their headaches stopped upon the use of medications or therapies. When asked students about their headaches being usually accompanied or preceded by vomiting, only 6.6% agreed that they lightly vomited, while most of the students (86.7%) disagreed. Similarly, when asked about their headaches being usually accompanied or preceded by nausea, only 19.5% reported that they had nausea (Table 3).

### 3.3. Management of Headache

Participants’ usage of medications and therapies for headaches is shown in Figure 3. Of the participants, 60.8% considered taking analgesics, while 26.5% considered sleeping, followed by caffeine consumption (14.4%) and herbal and alternative remedies (7.7%). However, as indicated in Figure 3, 17.1% of the individuals did not consider using any headache drugs.

### 3.4. Association between Pain Intensity and Student Characteristics

Table 4 describes the differences between pain intensity scores and participant characteristics (episodes of headache and year of study). According to the findings, the year of study has a significant difference with regard to headache pain intensity (*p* = 0.046), where fourth-year students were found to have higher headache pain intensity compared to other students (2.0645 ± 2.43; *p* = 0.046). Similarly, the pain intensity was significantly higher among students suffering from 1–3 episodes and >7 episodes of headaches (*p =* 0.001), as shown in Table 4.

### 3.5. Association between Participant Headache Episodes and Characteristics

The current findings showed that photophobia and phonophobia are not significantly associated with the episodes of headache per day (*p* = 0.116, *t* = 2.054, 95% *CI* = 0.01827–0.91685) (*p* = 0.257, *t* = 1.385, 95% *CI* = −0.14244–0.81320). Similarly, the family history of the students was also not significantly associated with the episodes of headache (*p* = 0.579, *t* = 0.934, 95% *CI* = −0.23376–0.65393). However, headache episodes were significantly associated with routine activities (reading writing, attending college; *p* = 0.018, t = 2.282) and daily activities (walking, running) (*p* = 0.022, *t* = 2.307). A detailed description is given in Table 5.

## 4. Discussion

The discussion of this study includes a summary of some assessment studies that have been conducted on the prevalence, frequencies, and characteristics of headaches both nationally and internationally [9,10,11,12]. This cross-sectional study was carried out among EMS students at a college connected with King Saud University in Riyadh, Saudi Arabia. According to our findings, 90.6% of the EMS students who took part in the survey reported at least one headache episode in the previous week. A study of female university students in Saudi Arabia found that 77% of them suffered headaches, with 58.4% having migraines and 41.6% having tension headaches [8]. In the regional context, a study conducted among health science university students of both sexes in India [20] and Ethiopia [36] found that headaches were prevalent in 63.9% and 67.22% of the participants, respectively [20,36]. The likely reasons for this discrepancy could be academic stress (studying for tests and/or examinations), altered sleeping patterns, absence of study breaks, and excessive workload.

The findings of the current study showed a negative attitude toward EMS students’ headaches and their routine activities, about 44.3% of students believed that headaches cause them to avoid routine activities. In contrast, other researchers reported that 71.3% of participants’ daily tasks were reduced due to the occurrence of headaches [37]. Conversely, a previous study by Menon and Kinnera in 2013 [38] reported 84% of students’ routine activities were aggravated by headaches. These findings were inconsistent with our findings where 40.3% of the EMS students in the current study agreed that their routine activities were aggravated by headaches. However, our findings were somewhat similar to previous findings by Bashatah et al., among pharmacy and nursing undergraduates, where the authors found that 34.7% of the participants were aggravated by headaches [12]. The current and previous findings indicate a heavy burden of stress, and it is important to identify the factors correlated with headaches and their aggravation.

A large proportion of EMS students in this study used analgesics and anti-inflammatories for treating headaches. However, our results were consistent with the other studies conducted by Ojini et al., among medical students at the University of Lagos, Nigeria [39]; Alkarrash et al., among healthcare students at the University of Aleppo [40]; and Al Hassan et al., among female students of King Faisal University, in Saudi Arabia [41], who reported that 68.2%, 66.4% and 64.9% of students used paracetamol for the management of headaches, respectively. In contrast, our study results were higher than those of a previous study conducted among female students in Taif (53.9%) and nursing students in India (36.76%) who preferred analgesics such as paracetamol for treating their headaches [8,20]. The most common reasons given as a justification to indulge in paracetamol, often recommended as one of the first treatments for headaches, are that it is one of the safest drugs and most widely used over-the-counter painkillers in the world and that side effects are rare [20,41]. Overall, the rate of using analgesics such as paracetamol for headaches observed here was similar to that in most of the studies conducted in Saudi Arabia and higher than that reported in other parts of the world. However, most of our students prefer sleeping and caffeine as other medications/therapies for headaches. Public health authorities such as the American Migraine Foundation and Medscape reported that good sleep often predicts relief from headaches. Outcome-based education could be an effective approach to bridging the gap between knowledge and practice [42,43].

The students’ low proclivity for any form of consultation was a noteworthy finding in our study; only 13.3% of them sought treatment from healthcare providers. A prior survey of medical students found that just 4.6% sought medical assistance [38]. A comparable study conducted among medical, dentistry, and pharmaceutical students at Aleppo University in Syria found that 15.9% sought medical advice [40]. A recent statewide survey of Polish cohorts, on the other hand, found that headache sufferers between the ages of 18 and 22 years waited an average of 2 years from the onset of symptoms to consult with medical specialists [44]. These data revealed that students and parents lacked knowledge regarding headache management and therapy [44], and it is worth noting that the student’s belief that headache is not a serious illness was also the main pretest finding among university students. An important point in our study was the students’ low tendencies to consult for medical assistance might be due to their belief that they were equipped with sufficient knowledge of self-medication to manage their headaches in addition to easy access to over-the-counter medicines. The majority (60.8%) of the study respondents had bilateral headaches, and this was also in line with the findings of the most comparable studies conducted by Bhat et al. among dental students (60.3%), Antonenko et al. among medical students (80.9%), and Bashatah et al. among pharmacy and nursing students (49.6%), which reported that respondents had a bilateral localization of their headaches [45,46].

In this study, 40.9% of the students indicated a throbbing/pulsating type of headache pain, followed by 52.5% with a pressing or tightening pain type, and 21% of them had sharp or stabbing headache pain. These findings were similar to previous findings presented by Bhat et al. and Bashatah et al. among healthcare students [12,45], of whom 61.9%, 22%, and 16.1% reported throbbing, pressing, and stabbing headache pain [12,45]. Similarly, another study reported that 23.7% of the students reported pressing headaches, 19.7% of the students reported stabbing headaches, and 23.7% of the students reported throbbing headaches [45]. It is perceived that students, particularly healthcare students, engage in many hours of reading, which may be a factor causing headaches among students [47,48].

The most common risk factors for headaches among students were anxiety, stress from their education, a lack of proper vacation time, and a family history of headaches [47,48]. Lack of sleep, intellectual exertion, and stress were also reported in other studies [48,49]. Similarly, Panigrahi et al. discovered that being a female, difficulty going to sleep, difficulty staying asleep, soft drink intake, and self-dissatisfaction with one’s health were all connected with headache [10]. As a result, it is crucial to address these factors by quick monitoring, and identifying triggering factors is an important step in reducing the frequency and severity of headaches and related difficulties. As a result, academics, health professionals, and all other interested parties must dedicate sufficient attention to developing headache prevention and treatment strategies.

In this study headache episodes had a significant association with routine activities (reading, writing, attending college) (*p* = 0.018, *t* = 2.282) and everyday activities (walking, jogging) (*p* = 0.022, *t* = 2.307). Furthermore, when compared to other students, fourth-year EMS students had greater headache pain intensity scores (*p* = 0.046). Similarly, the severity of discomfort was considerably higher among students who had 1–3 and >7 occurrences of headaches (*p* = 0.001). While a comparable study found that students’ age, gender, course of study, smoking habit, and kind of headache were all associated with headache intensity, headache intensity was also associated with routine activity (*p* = 0.0001). [12]. This indicates that undergraduates suffer from headaches, which may occur due to a variety of factors, factors related to learning to load in their academics, burden of exams and presentations, which may lead to sleeplessness and stress full days and nights. There may be certain limitations to our study. The study was conducted at a single college in a single university in Riyadh, Saudi Arabia, so the results may not accurately reflect the headaches posed by all healthcare undergraduates in Saudi Arabia or other countries. Additionally, the data included in this study were restricted to only emergency medical services students. Third, as women are still not allowed to attend college alongside men in Saudi Arabia and because both male and female colleges have separate campuses, the survey did not include women because there were few responses and therefore excluded from the study. Although policymakers may be able to perform a timely assessment of social issues using the findings of this kind of assessment and make recommendations as a result.

## 5. Conclusions

Given the study’s findings, 90.6% of the students suffered from at least one episode of headache suggesting that headaches are the most frequently occurring neurological condition that disturbs their academic life and further leads to stressful days and nights. The current findings also revealed that headaches interfere with daily activities, which may lead to absenteeism, from college, and lower students’ grades or even cause them to fail in their academic careers. The analgesics, sleeping, and use of caffeine were the most common management approaches among the students. We advise headache sufferers to always try to stay hydrated, use relaxation techniques like head massage and hot or cold compresses to the head or neck, avoid mobile devices and screens to give their eyes breaks, rest in a quiet place, and recognize and avoid the foods that cause headaches for the best headache control. Additionally, having adequate sleep, use of little caffeine are other best ways to control the headache and its severity. Consequently, we advocate for the implementation of educational initiatives that would educate students about headache prevention and treatment.

## Figures and Tables

**Figure 1 medicina-59-01522-f001:**
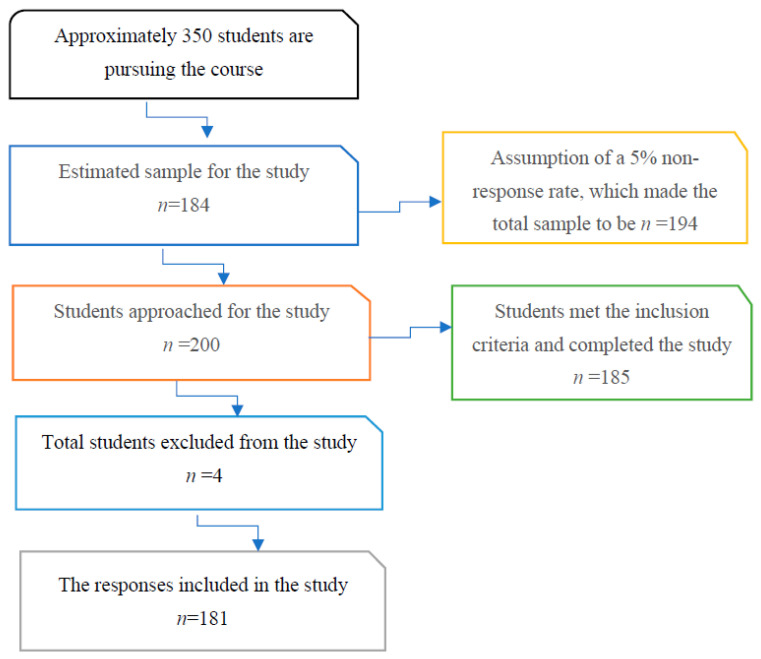
Flowchart of the responses.

**Figure 2 medicina-59-01522-f002:**
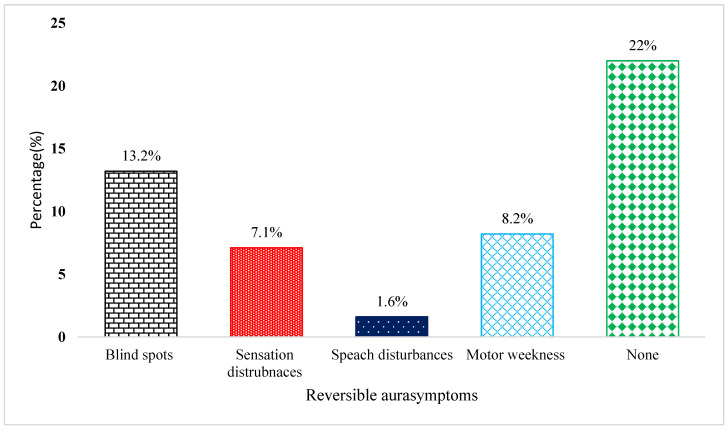
Symptoms that are accompanied or followed by headache episodes.

**Figure 3 medicina-59-01522-f003:**
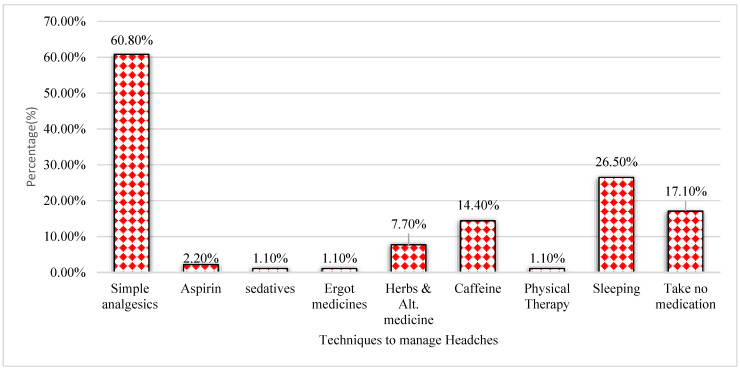
Medications/therapies used for headaches.

**Table 1 medicina-59-01522-t001:** Demographic and clinical characteristics of undergraduates (*n* = 181).

Parameters	Frequency (n)	Percentage (%)
Year of Study 2nd year 3rd year 4th year Final year	70 64 31 16	38.7% 35.4% 17.1% 8.8%
Marital status Married Single/unmarried	4 177	2.2% 97.8%
Have you experienced at least one headache in the recent week? Yes No	164 17	90.6% 9.4%
How many days have you had a headache in the last week? 0-days 1-day 2-days 3-days 4-days 5-days 6-days 7-days	22 34 39 28 19 20 08 11	12.2% 18.8% 21.5% 15.5% 10.5% 11.0% 4.4% 6.1%
Concerning the previous question, how many episodes do you have per day? At least one episode Between 1 and 3 episodes Between 4 and 6 episodes >7 episodes	33 134 12 2	18.2% 74% 6.6% 1.1%
The duration of each episode (in hours/seconds) * 7–30 min 40–60 min 1–4 h 5–6 h	3 33 26 103	1.6% 18.1% 14.3% 56.6%

* = missing response; n = frequency; % = percentile.

**Table 2 medicina-59-01522-t002:** Headache characteristics among participants.

Variables	Frequency *n* (%)
The site of the headache Frontal Occipital Vertical Temporal Generalized Orbital	88 (48.6%) 31 (17.1%) 16 (8.8%) 49 (27.1%) 21 (11.6%) 42 (23.2%)
Your headache is Bilateral Unilateral	110 (60.8%) 71 (39.2%)
Characteristics of the headache pain Throbbing/pulsating Pressing/tightening Sharp/stabbing	74 (40.9%) 95 (52.5%) 38 (21.0%)
Do routine academic activities (like going to college, reading, and writing) increase your headache? Yes No	73 (40.3%) 108 (59.7%)
Do you avoid normal activities (walking, running) because of your headache? Yes No	80 (44.2%) 101 (55.8%)

*n* = frequency; % = percentile.

**Table 3 medicina-59-01522-t003:** The intensity of pain and other symptoms associated with headache and management approaches among healthcare students.

Variables	Mean ± SD	Frequency *n* (%)
How severe is your headache on average? (0 represents no pain and 10 represents the worst pain)	5.016 ± 1.92 (median = 5)	
Is it common for your headache to be accompanied or preceded by vomiting? Yes No		16 (8.8%) 165 (91.2%)
The severity of your vomiting No vomiting Lightly vomited Moderately vomited Strongly vomited Severely vomited		157 (86.7%) 12 (6.6%) 8 (4.4%) 2 (1.1%) 2 (1.1%)
Is your headache generally preceded by nausea? Yes No		36 (19.9%) 145 (80.1%)
The intensity of your nausea No nausea Light nausea Moderate nausea Strong nausea Severe nausea		146 (80.5%) 26 (14.4%) 5 (2.8%) 3 (1.7%) 1 (0.6%)
Is your headache generally accompanied by light sensitivity? Yes No		61 (33.7%) 120 (66.3%)
Is your headache frequently accompanied by a fear of loud sounds? Yes No		50 (27.6%) 131 (72.4%)
Do you have a family history of headaches? Yes No		69 (38.1%) 112 (61.9%)
Have you ever sought medical help for a headache? (For example, a doctor’s appointment) Yes No		24 (13.3%) 157 (86.7%)
Does your headache go away after you take your medication? Yes No I take no medications for my headache.		116 (64.1%) 29 (16.0%) 36 (19.9%)

*n* = frequency; % = percentile.

**Table 4 medicina-59-01522-t004:** Differences between pain intensity scores and participant year for study headache episodes per day.

Variables	Mean	Std. Deviation	Std. Error	95% Confidence Interval for Mean		*p*-Value *
				Lower Bound	Upper Bound	
Year of Study						
Second year	1.2571	1.08595	0.12980	0.9982	1.5161	0.046
Third year	1.5156	1.25978	0.15747	1.2009	1.8303	
Fourth year	2.0645	2.43496	0.43733	1.1714	2.9577	
Final year	1.0625	0.68007	0.17002	0.7001	1.4249	
Episodes per day						
At least one episode	3.8125	2.23517	0.39513	3.0066	4.6184	0.001
Between 1 and 3 episodes	5.3134	1.77458	0.15330	5.0102	5.6167	
Between 4 and 6 episodes	4.9167	1.67649	0.48396	3.8515	5.9819	
>7 episodes	5.0000	0.00000	0.00000	5.0000	5.0000	

* One-way ANOVA.

**Table 5 medicina-59-01522-t005:** Differences between participant headache episodes and characteristics.

Variables	Mean (Std)	Standard Error Mean (SEM)	*t*	*p*-Value *
Have you ever sought medical attention for your headache? Yes No	1.70 (1.122) 1.45 (1.504)	0.22904 0.12081	0.802	0.641
Does your headache aggravate routine academic activities? Yes No	1.76 (1.837) 1.26 (1.098)	0.21504 0.10574	2.282	0.018
Does your headache cause you to avoid daily activities (walking, running)? Yes No	1.76 (1.77) 1.26 (1.10)	0.19829 0.11083	2.307	0.022

* Independent sample *t*-test.

## Data Availability

The data used for this study will be made available from the corresponding author upon request.

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
