# Peer review of "Evaluation of Headache Trends among Undergraduate First Responders for Medical Emergencies at Saudi University in Riyadh, Saudi Arabia"

_medicina, 2023, doi:10.3390/medicina59091522_

Round 1

Reviewer 1 Report

This is a cross-sectional prospective analysis of data collected among male students. Unfortunately, authors present little convincing data that this particular cohort requires evaluation in the context of headache prevalence. Why future paramedics are potentially more prone to headache disorders and which types?

Major drawbacks of this study include:

- No ethical committee approval. Or at least no declaration on how this issue was addressed.

- Most respondents were 18-22 years old. According to a recent large study in a migraine cohort the this is the approximate time when first migraine attacks occurred in the majority of patients (mean age 19.17, median 18). In that study patients waited on average 2 years form first attacks to consulting with medical specialist. This can further explain why subjects in your cohort did not yet consulted their headache with a physician. (doi: 10.1186/s10194-023-01575-4)

- There seems to be inconsistency in the collected data that authors should address - 18% respondents reported zero headache episodes per day, while at the same time only 12% of respondents declined having any headaches in the previous week. That leaves 6% of respondents who had zero headache episodes but had a headache in a previous week.

- It seems that authors collected the relevant data to attribute the headache at least to some primary headache diagnoses according to International Classification of Headache Disorders – 3 (migraine without aura and perhaps tension-type headache). I recommend additional analysis that would improve the quality of the study. However, this may be obstructed by question regarding headache attack duration that is incompatible with any primary headache diagnoses (max 6 hours).

English requires reediting by native speaker with experience in medical writing. E.g., in the title authors use phrase: „undergraduates of first responders for medical emergencies” which is somehow unclear and requires editing. 

Author Response

Reviewer 1

Comments and Suggestions for Authors

This is a cross-sectional prospective analysis of data collected among male students. Unfortunately, authors present little convincing data that this particular cohort requires evaluation in the context of headache prevalence. Why future paramedics are potentially more prone to headache disorders and which types?

Response: Dear editor and team of review very good morning, we highly appreciate your valuable time to review and your potential comments, since in ksa there were studies examining the headache trends among health care professionals, students (medical, pharmacy) and public, but no study assessed the headache among EMS students.

This assessment is cortical, since Ems students are future practising professionals. Furthermore, EMS students are one of the most valuable healthcare professionals, responsible for safely and quickly diagnosing and managing in emergencies or at the time of any disasters, they are required to act quickly and efficiently to save lives and minimize injuries. These data can be used to assess whether or not students believe that parts of their education cause their headaches, as well as how much they believe headaches negatively affect their normal activities (Going to college, walking) routine activities, (e.g., reading, writing, labs,). Therefore, there is a need to investigate the Headache Trends among undergraduates of first responders for medical emergencies at Saudi University in Riyadh, Saudi Arabia. Furthermore, we did not look in to the type of headache but we assessed the severity of pain, how frequent is the headache, and overall are the students suffering with headache or not?

Major drawbacks of this study include:

- No ethical committee approval. Or at least no declaration on how this issue was addressed.

Response: Dear team of reviewers very good morning, we highly appreciate your valuable time to review and your potential comments,

we apologise for the missing information, of course this study was approved by research ethics comments college of medicine king Saud university, Riyadh ‘Saudi Arabia

- Most respondents were 18-22 years old. According to a recent large study in a migraine cohort the this is the approximate time when first migraine attacks occurred in the majority of patients (mean age 19.17, median 18). In that study patients waited on average 2 years form first attacks to consulting with medical specialist. This can further explain why subjects in your cohort did not yet consulted their headache with a physician. (doi: 10.1186/s10194-023-01575-4).

Response: thank you so much for the valuable suggestion, we agree with you, the current findings included young students aged between 18-22 years. I added the explanation as follows. On the other hand, a recent nationwide survey among polish cohorts reported that headache suffers with young ages between 18-22 years waited on average 2 years form first attacks to consulting with medical specialist [44]. These findings and demonstrated that students and parents were lacking the education about the management and treatment of Headache. [44] or 

- There seems to be inconsistency in the collected data that authors should address - 18% respondents reported zero headache episodes per day, while at the same time only 12% of respondents declined having any headaches in the previous week. That leaves 6% of respondents who had zero headache episodes but had a headache in a previous week.

response: we apologise for the confusion; we appreciate your potential insights. This study included frequent headache sufferers as one of the inclusion criteria and 18% of them reported at least one episodes of headache in the past week, however 12.2% of them not reported having headache in the last week, they may have had headache after that week, but in the last week

- It seems that authors collected the relevant data to attribute the headache at least to some primary headache diagnoses according to International Classification of Headache Disorders – 3 (migraine without aura and perhaps tension-type headache). I recommend additional analysis that would improve the quality of the study. However, this may be obstructed by question regarding headache attack duration that is incompatible with any primary headache diagnoses (max 6 hours).

response: we apologise for the confusion, as we seen from the results, and we can say that Chronic tension-type headaches were contributed to 56.6% of the population, while rest of them were recognized as a primary headache. Although our aim is to assess the headache and their trends, like frequency of headache, pain intensity, severity, causes of headache, management, not the type of headache. However, we have run additional analysis as suggested

Comments on the Quality of English Language

English requires reediting by native speaker with experience in medical writing. E.g., in the title authors use phrase: „undergraduates of first responders for medical emergencies” which is somehow unclear and requires editing. 

response: We appreciate your valuable comments and the entire manuscript was revised for English editing

Reviewer 2 Report

Comments:

This is a study about evaluating headache trends among undergraduates of medical emergency in a Saudi university. The study question is neither novel, nor interesting. Following points needs to be addressed:

1-      The research question is not novel. Similar studies has been done in many different countries. Authors have not highlighted any existing knowledge gap to justify thus study. Stress, pressure to perform, long working hours are also seen with other professions. Why the authors hypothesize the headache in EMS students will be different? Authors need to build a hypothesis for their research question.

2-      The study is a replication of multiple previous studies with no novelty.

3-      I wonder, why the authors did not attempt to diagnose the type of headache as per ICHD3 ? This would have helped to characterize different type of headache disorder.

4-      The inclusion criteria states, students with age >18 yrs, could speak Arabic/English, providing consent. 181 participants responded. The headache characteristics (table 2) is of 181 participants. So all the participants had headache, this makes the prevalence as 100% !. It must be stated that, inclusion criteria does not mention about presence of headache as one of the criteria.

5-      If the authors intended to evaluate the headache in EMS residents, then questions should have been tailored for them. Temporal association with night duties, prolonged duties, skipping of meals etc should have been included.

6-      In Table 1, for the question “how many episodes do you have per day” 33 (18.2%) responded as zero. But for the question.. ”How many days you had a headache in last  week”  only 22 (12.2%) responded as zero. The discrepancy may be explained.

7-      A patient reported headache impact scale could have been used to assess the overall impact of headache on quality of life.

8-      Important information e.g duration of headache, age of onset of headache should have been collected.

9-      The analysis in table 4 is devoid of any logical hypothesis. Pain intensity is a subjective score and it varies from episodes to episodes in a particular person. Hence to evaluate the factors for pain intensity is devoid of any scientific basis.

10-  How the intensity of nausea was categorised as “light, moderate, strong, severe”?

Minor editing of English language is required. 

Author Response

Comments:

This is a study about evaluating headache trends among undergraduates of medical emergency in a Saudi university. The study question is neither novel, nor interesting. Following points needs to be addressed:

Response:  Firstly of all my heart fully thanks to the reviewers for their valuable time spent during this review process, we sic nearly apologies for the any errors and we followed all your recommendations to improve the manuscript. Furthermore, this study questionnaire was used by other population, which includes, medicals students, nursing and pharmacy’s students  

1-      The research question is not novel. Similar studies have been done in many different countries. Authors have not highlighted any existing knowledge gap to justify thus study. Stress, pressure to perform, long working hours are also seen with other professions. Why the authors hypothesize the headache in EMS students will be different? Authors need to build a hypothesis for their research question.

Response: we appreciate your comments, and we agreed that similar studies done elsewhere, of course but not in the similar population what we studied (ems) , since ems students are potential primary responders during the emergencies , their knowledge and their degree would not only helps during their practice site , but also helps in providing adequate care and health outcomes among public.

This study is unique, since no single study assessed that headaches (primary headaches) their characters and managing techniques among EMS.

Since ems professionals are primary responders in emergency, comparison to other professionals, they have to responded very fast in comparing to others, their presence of good health provides adequate treatment and outcomes,

2-      The study is a replication of multiple previous studies with no novelty.

response: we appreciate your comment , and we agree that similar studies exist among pharmacy AND medical professionals , of course as of now there were no single study published in Saudi Arabia or even in other countries to assess the headache trends among population of first responders, since they deal with emergency situation , either natural incidence , or manmade incidence , if they are not active or not healthy , how they deal in emergency situation , and they are the one more prone to hatches , due to pressure in work , nature of their degree, labs &practical’s , and internships

3-      I wonder, why the authors did not attempt to diagnose the type of headache as per ICHD3? This would have helped to characterize different type of headache disorder.

Response: we appreciate your comment, but this study aimed to assess the frequency of headache and its characters, not the type of headache, further if we see the data we can conclude that there were primary headaches and Chronic tension-type headaches were contributed to 56.6%.

4-      The inclusion criteria states, students with age >18 yrs, could speak Arabic/English, providing consent. 181 participants responded. The headache characteristics (table 2) is of 181 participants. So all the participants had headache, this makes the prevalence as 100% !. It must be stated that, inclusion criteria do not mention about presence of headache as one of the criteria.

Response: we appreciate your comment, and we apologies for the confusion. The target population consisted of all students over the age of 18 who commonly suffered from headaches, could speak Arabic or English, and accepted to participate in the study by signing a consent form. Students with headaches caused by the virus, hangover, cold, or head injury, as well as those who did not meet the inclusion requirements, were eliminated from the study. 

5-      If the authors intended to evaluate the headache in EMS residents, then questions should have been tailored for them. Temporal association with night duties, prolonged duties, skipping of meals etc. should have been included.

Response: we appreciate your comment, this study was conducted among students, we aimed to assess the headache trends not the factors that causes headache, however, according to literature, headache among students might be cause of academic elated stress, and preparing for emergencies among EMS, and I have cited this as … Temporal association with night duties, prolonged duties, skipping of meals etc. should have been included in the discussion

6-      In Table 1, for the question “how many episodes do you have per day” 33 (18.2%) responded as zero. But for the question.” How many days you had a headache in last week” only 22 (12.2%) responded as zero. The discrepancy may be explained.

response: we apologies for the confusion and we appreciate your question, it was a typo error. because we assessed the data about last week, but the 12.2% they did not have in the last week but after that week they reported headache

7-      A patient reported headache impact scale could have been used to assess the overall impact of headache on quality of life.

response: we apologies, and we aprerecit your comments and in the future studies we will consider to assess the quality of life as well

8-      Important information e.g. duration of headache, age of onset of headache should have been collected.

Response: we apologies, and we appreciate your comments, the duration is mentioned in the table-1 and age too as follows. The mean age of the students suffered with headache was 20.40 (median age was 20.0)

Figure onset of age for the headache

9-      The analysis in table 4 is devoid of any logical hypothesis. Pain intensity is a subjective score and it varies from episodes to episodes in a particular person. Hence to evaluate the factors for pain intensity is devoid of any scientific basis.

Response: We appreciate your comments and we apologies for this errors, we removed that table which describes the differences between pain intensity scores and participant headache characteristics. And furthermore we did a new analysis about differences between pain intensity scores and participant headache episodes as well as year of study. The findings reported that fourth year EMS students found to have higher headache pain intensity’s cores comparing to other students (p=0.046). Similarly, the pain intensity was significantly higher among the students between 1-3 and >7 episodes of headaches(p=0.001) as shown in the table-4

Table 4. describes the differences between pain intensity and participant headache episodes.

Variables

Mean

Std. Deviation

Std. Error

95% Confidence Interval for Mean

p-value *

Lower Bound

Upper Bound

Year of study

Second year

1.2571

1.08595

.12980

.9982

1.5161

0.046

Third year

1.5156

1.25978

.15747

1.2009

1.8303

Fourth year

2.0645

2.43496

.43733

1.1714

2.9577

Final year

1.0625

.68007

.17002

.7001

1.4249

Episodes per day

At least one episode

3.8125

2.23517

.39513

3.0066

4.6184

0.001

Between 1-3 episodes

5.3134

1.77458

.15330

5.0102

5.6167

Between 4-6 episodes

4.9167

1.67649

.48396

3.8515

5.9819

>7 episodes

5.0000

.00000

.00000

5.0000

5.0000

*one-way ANOVA

Table 5. describes the differences between participant headache episodes and characters

Variables

Mean (Std)

SEM

t

p-value*

Have you ever sought medical attention for your headache?

Yes

No

1.70(1.122)

1.45(1.504)

0.22904

0.12081

0.802

0.641

Does your headache aggravate with routine academic activities?

Yes

No

1.76(1.837)

1.26(1.098)

0.21504

0.10574

2.282

0.018

Does your headache cause you to avoid daily activities (walking, running)?

Yes

No

1.76(1.77)

1.26(1.10)

0.19829

0.11083

2.307

0.022

*Independent sample t-test.

Although the current finding’s reported that photophobia and phono phobia is not significantly associated with the episodes of headache per day (p=0.116 t=2.054; 95% CI = .01827-.91685); (P=0.257 T=1.385 95% CI= -.14244-.81320). Similarly, the family history of the students also not significantly associated with the episodes of headache (p=0.579; t=0.934; 95% CI= - 0.23376 - 0.65393). However, headache episodes were significantly associated with routine activates (reading writing, attending college,) (p=0.018 t=2.282) daily activities (walking, running) (p=0.022 t=2.307)

10-  How the intensity of nausea was categorized as “light, moderate, strong, severe”?

Response:  thank you for the comment and in this study … light nausea is defined as urge to vomit (Not all people who feel nauseated throw up, but many have the overwhelming sensation that throwing up would help them feel better.)

moderate nausea is accompanied by sweating, weakness, increased saliva, and an urge to vomit

strong nausea, has left you unable to eat or drink for more than 12 hours this type of nausea required pharmacological medication to treat. While the severe nausea is doesn't subside within 24 hours of trying over-the-counter interventions. and resistant to required pharmacological medication

Comments on the Quality of English Language

Minor editing of English language is required. 

Response:  thank you for the comment, the entire manuscript was revised for English language

Round 2

Reviewer 1 Report

I appreciate the authors response. However I can see little changes to the manuscript that acknowledge my notes (i.e. no ethical disclaimer, explanation of data inconsistency, calculations of prevalence of migraine  or tension-type headache).

no additional comments

Author Response

Author's Reply to the Review Report (Reviewer 1)

Comments and Suggestions for Authors

I appreciate the authors response. However, I can see little changes to the manuscript that acknowledge my notes (i.e. no ethical disclaimer, explanation of data inconsistency, calculations of prevalence of migraine or tension-type headache).

Response: Respected reviewers and team of editors, firstly my sincere thanks and we appreciate your valuable comments for the improvement of our research

we sincerely apologies for the missing information,

This study as approved by ethical committee college of medicine at King Saud University (IRB Project No. E-23-8027), prior to data collection, furthermore the informed consent was obtained from all the participants and they were informed that the data provided would be used only for the research purpose.

Furthermore, a clear picture of the participant responses was given in the Figure-1

We calculated the prevalence of tension-type headache based on the ICHD3 CRITERIA

At least 1-10 episodes of headache occurring on <1 DAY/Month an average (<12 days / year) and lasting from 30-minutes to 7 days

At least two of the following four characteristics:

bilateral location 2. pressing or tightening (non-pulsating) quality 3. mild or moderate intensity 4. not aggravated by routine physical activity such as walking or climbing stairs

Both of the following: 1. no nausea or vomiting 2. no more than one of photophobia or photophobia

Reviewer 2 Report

Comments:

The responses given to my previous queries are not adequate and satisfactory. The study design is flawed and high probability of selection bias is noted. The statistical analysis is not based on any logical hypothesis and the representation of data is poor.  The responses provided by the authors have led to new questions:

1-     In response to my query no 3, the authors replied that they did not intended to classify the headache. However, in the next sentence, they have written that…” primary and chronic tension type headache were contributed to 56.6%”.  Now, how the authors diagnosed the tension type headaches? Which criteria dis they follow?

2-     There is a mismatch of data in First paragraph of results section and table 1. In text, it is written 56.6% (n=103) had headache lasting between 1-4 hrs, but in Table 1, the same has been mentioned as 5-6 hrs.

3-     In response to query 4, authors mention that they targeted students who suffered from headaches. Now how did they screen patients with headache? This introduces a major selection bias in the study. If authors intended to include only students with headache, then presence of headache should have been an inclusion criterion.

4-     In response to query 8, authors mention the age as 20.4 . But it is not mentioned in the manuscript. Moreover, my query was “duration of headache (as a disease)”. Age of onset of headache (not the age of the study population)

5-     The new analysis in Table 4 is again devoid of any logical hypothesis. The representation of data in the table is also confusing. What does the mean for the row “second year”  (1.2571) represent? Is it headache frequency or pain intensity? This is very poor representation of data.

6-     In response to query 10, authors have provided definition of light/moderate/strong nausea. I am curious to know the reference for the same. The same classification/definition has not been provided in the revised manuscript. Moreover, the question of categorizing nausea in relation to headache is illogical. A single patient can have different degrees of nausea in different headache episodes. To combine them in to groups and analyzing for research hypothesis is not scientific and highly likely to lead to erroneous conclusions.

Minor editing of english language is required.

Author Response

Comments:

The responses given to my previous queries are not adequate and satisfactory. The study design is flawed and high probability of selection bias is noted. The statistical analysis is not based on any logical hypothesis and the representation of data is poor.  The responses provided by the authors have led to new questions:

1-     In response to my query no 3, the authors replied that they did not intended to classify the headache. However, in the next sentence, they have written that…” primary and chronic tension type headache were contributed to 56.6%”.  Now, how the authors diagnosed the tension type headaches? Which criteria dis they follow?

Response: We sincerely apologies for the confusion, as we told earlier we have not looked in to type of headache, our aim was to assess the headache trends (characters, frequency, pain intensity, academic life with headache , managing techniques )  however after your recommendation that it would be good to classify headache types and we have gone through the International Classification of Headache Disorders 3 rd. Edition (ICHD-3), according to following criteria , in our study 56.6% of the students suffering with tension type of headache as follows

We calculated the prevalence of tension-type headache based on the ICHD3 CRITERIA

At least 1-10 episodes of headache occurring on <1 DAY/Month an average (<12 days / year) and lasting from 30-minutes to 7 days

At least two of the following four characteristics:

bilateral location 2. pressing or tightening (non-pulsating) quality 3. mild or moderate intensity 4. not aggravated by routine physical activity such as walking or climbing stairs

Both of the following: 1. no nausea or vomiting 2. no more than one of photophobia or photophobia

2-     There is a mismatch of data in First paragraph of results section and table 1. In text, it is written 56.6% (n=103) had headache lasting between 1-4 hrs., but in Table 1, the same has been mentioned as 5-6 hrs.

Response: We apologies for the typo error, since the amsnucirpt was written by all of us 3 authors and sometimes, typological errors contribute in this stage, we appreciate your comment and we corrected the manuscript. as follows. Furthermore, 56.6%(n=103) of the students reported 5-6 hours, while 14.3%(n-26) of the students reported that their headache last between 1-4 hours.

3-     In response to query 4, authors mention that they targeted students who suffered from headaches. Now how did they screen patients with headache? This introduces a major selection bias in the study. If authors intended to include only students with headache, then presence of headache should have been an inclusion criterion.

Response: Dear reviewer, it is an excellent question for us, we agreed with you and it is already there in the manuscript as a one of the inclusion criteria and I have highlighted it now.

Furthermore, before carrying out the study we asked the students to come forward for answering the survey, only those students who are frequently suffering with the headache, in the current week or past week. We included this question in the questionnaires. Therefore, we approached 200 students, but only 181 met the inclusion criteria and eligible for the study m, therefore we included only 181 students …

The target population consisted of all students over the age of 18 who commonly suffered from headaches, could speak Arabic or English, and accepted to participate in the study by signing a consent form. Students with headaches caused by the virus, hangovers, colds, or head injuries, as well as those who did not meet the inclusion requirements, were eliminated from the study.

4-     In response to query 8, authors mention the age as 20.4. But it is not mentioned in the manuscript. Moreover, my query was “duration of headache (as a disease)”. Age of onset of headache (not the age of the study population)

Response: Dear reviewer, thank you for the comment, we apologies for this issue, we thought the mean age of the respondents suffering with headache, therefore we calculated, and we already have the age in continuous scale, the duration of headache we calculated as follows and mentioned in the manuscript in table 1

5-     The new analysis in Table 4 is again devoid of any logical hypothesis. The representation of data in the table is also confusing. What does the mean for the row “second year” (1.2571) represent? Is it headache frequency or pain intensity? This is very poor representation of data.

Response: We apologies for your confusion. Thank you for the comment respected reviewer, we appreciate it. we corrected the table with more clear explanation.  as follows According to the findings, the year of study has a significant difference with regards to headache pain intensity (p= 0.046), where fourth-year students were found to have higher headache pain intensity compared to other students (2.0645 ±2.43; p=0.046). Similarly, the pain intensity was significantly higher among students suffering with 1-3 episodes and >7 episodes of headaches (p=0.001) as shown in Table 4

6-     In response to query 10, authors have provided definition of light/moderate/strong nausea. I am curious to know the reference for the same. The same classification/definition has not been provided in the revised manuscript. Moreover, the question of categorizing nausea in relation to headache is illogical. A single patient can have different degrees of nausea in different headache episodes. To combine them in to groups and analyzing for research hypothesis is not scientific and highly likely to lead to erroneous conclusions.

Response: thank you for the comment, we agree with you, but when we ask the students about the question, they answered LIGHT, MODERATE, STRONG, AND SEVERE,

although in this study majority 80.7% did not felt nausea (therefore we can say that 80.75 of the students felt no nausea

light nausea is defined as urge to vomit (Not all people who feel nauseated throw up, but many have the overwhelming sensation that throwing up would help them feel better.)

moderate nausea is accompanied by sweating, weakness, increased saliva, and an urge to vomit

strong nausea, has left you unable to eat or drink for more than 12 hours this type of nausea required pharmacological medication to treat. While the severe nausea is doesn't subside within 24 hours of trying over-the-counter interventions. and resistant to required pharmacological medication

References: https://www.google.com/url?sa=t&rct=j&q=&esrc=s&source=web&cd=&ved=2ahUKEwjnkO7cj6eAAxV1V6QEHU6HBGoQFnoECA8QAw&url=https%3A%2F%2Fwww.everydayhealth.com%2Fnausea%2Fguide%2F%23%3A~%3Atext%3DNausea%2520is%2520the%2520feeling%2520of%2Ccauses%2520of%2520nausea%2520and%2520vomiting.&usg=AOvVaw0yeBPqvMBLTgRxhU60pCuC&opi=89978449.

Kjeldgaard HK, Vikanes Å, Benth JŠ, Junge C, Garthus-Niegel S, Eberhard-Gran M. The association between the degree of nausea in pregnancy and subsequent posttraumatic stress. Arch Womens Ment Health. 2019 Aug;22(4):493-501. doi: 10.1007/s00737-018-0909-z. Epub 2018 Sep 17. PMID: 30225528; PMCID: PMC6647437.